# Crude α-Mangostin Suppresses the Development of Atherosclerotic Lesions in *Apoe*-Deficient Mice by a Possible M2 Macrophage-Mediated Mechanism

**DOI:** 10.3390/ijms20071722

**Published:** 2019-04-07

**Authors:** Masa-Aki Shibata, Mariko Harada-Shiba, Eiko Shibata, Hideki Tosa, Yoshinobu Matoba, Hitomi Hamaoka, Munekazu Iinuma, Yoichi Kondo

**Affiliations:** 1Department of Anatomy and Cell Biology, Osaka Medical College, 2-7 Daigaku-machi, Takatsuki, Osaka 569-8686, Japan; an1021@osaka-med.ac.jp (H.H.); konchan@osaka-med.ac.jp (Y.K.); 2Department of Molecular Innovation in Lipidology, National Cerebral & Cardiovascular Center Research Institute, Suita, Osaka 565-8565, Japan; mshiba@ncvc.go.jp (M.H.-S.); belleiko@gmail.com (E.S.); 3Field & Device Co., Osaka 541-0045, Japan; h-tosa@field-and-device.jp; 4Ecoresource Institute Co., Ltd., Minokamo, Gifu 505-0042, Japan; y.matoba429@kcn.jp; 5Gifu Pharmaceutical University, Gifu 502-8585, Japan; iinumamunekazu@gmail.com

**Keywords:** atherosclerosis, fatty liver, xanthones, macrophages, M1, M2, therapeutic, *Apoe* knockout mouse

## Abstract

Lifestyle choices play a significant role in the etiology of atherosclerosis. Male *Apoe*^−/−^ mice that develop spontaneous atherosclerotic lesions were fed 0%, 0.3%, and 0.4% mangosteen extracts, composed largely of α-mangostin (MG), for 17 weeks. Body weight gains were significantly decreased in both MG-treated groups compared to the control, but the general condition remained good throughout the study. The levels of total cholesterol (decreased very-low-density lipoprotein in lipoprotein profile) and triglycerides decreased significantly in the MG-treated mice in conjunction with decreased hepatic *HMG-CoA* synthase and *Fatty acid transporter*. Additionally, increased serum lipoprotein lipase activity and histopathology further showed a significant reduction in atherosclerotic lesions at both levels of MG exposure. Real-time PCR analysis for macrophage indicators showed a significant elevation in the levels of *Cd163*, an M2 macrophage marker, in the lesions of mice receiving 0.4% MG. However, the levels of *Nos2*, associated with M1 macrophages, showed no change. In addition, quantitative immunohistochemical analysis of macrophage subtypes showed a tendency for increased M2 populations (CD68^+^/CD163^+^) in the lesions of mice given 0.4% MG. In further analysis of the cytokine-polarizing macrophage subtypes, the levels of *Interleukin13* (*Il13*), associated with M2 polarization, were significantly elevated in lesions exposed to 0.4% MG. Thus, MG could suppress the development of atherosclerosis in *Apoe*^−/−^ mice, possibly through an M2 macrophage-mediated mechanism.

## 1. Introduction

According to the World Health Organization, ischemic heart disease and stroke are the world’s biggest killers, accounting for a combined 15 million deaths in 2015 [1]. These diseases have remained the leading causes of death globally in the last 15 years [1], and the primary etiologic factor in both conditions is atherosclerosis, which causes vessel stenosis, embolus via thrombus formation, and myocardial and cerebral ischemia [2].

The pericarp of the mangosteen, *Garcinia*
*mangostana* Linn (Figure 1A), has a long history of use as a medicinal plant in Southeast Asia [3]. When damaged, the pericarp secretes a yellow substance (Figure 1B) that functions to protect the fruit from bacterial infection and acts as an insect repellant [4]. This yellow exudate contains a class of compounds called xanthones, which includes α-, β-, and γ-mangostin (MG) (Figure 1C,D) [3]. Xanthones are also well-known anti-inflammatory agents, and, in fact, the mangosteen pericarp has long been widely used as an anti-inflammatory agent in Southeast Asia [3]. The anti-tumorigenic actions of pure α-MG and crude MG have also been shown in various type of cancer cells [5,6] and animal models for cancers of the mammary gland [7,8,9], colon [10,11], and prostate [12]. More recently, treatment with synthetic α-MG dilaurate, prepared by adding lauric acid, a medium-chain fatty acid with a high propensity for lymphatic absorption, to α-MG, resulted in strong suppression of wide-spectrum organ metastasis in a mouse model of mammary cancer [9].

α-MG is reported to reduce low-density lipoprotein (LDL) oxidation [13]. In addition, α-MG has been shown to attenuate lipid peroxidation [14]. Furthermore, in an in vitro study, α- and γ-MG inhibited nuclear factor-κB activity [15] and suppressed lipopolysaccharide-upregulated cyclooxygenase 2 and prostaglandin E2 (PGE2) in C6 rat glioma cells [16]. Both α- and γ-MG have also been shown to inhibit nitric oxide and PGE2 production in lipopolysaccharide-stimulated murine macrophage RAW 264.7 cells [17]. In a human study, daily consumption of a mangosteen-based formula for 30 days significantly reduced C-reactive protein without any side effects [18].

Since these actions by MGs are believed to induce suppressive effects on the development of atherosclerosis, we investigated whether crude α-MG, comprising 84% α-MG and 7% γ-MG, would inhibit the development of atherosclerotic lesions in *Apoe*-deficient (*Apoe*^−/−^) mice. Furthermore, possible mechanisms of the atherosclerotic effects of MG were explored.

## 2. Results

### 2.1. Survival and Body Weight

A single mouse receiving 0.4% MG died due to malocclusion at Week 10 and was excluded from analysis. Changes in body weight of control and MG-treated mice are shown in Figure 2A,B (0% vs. 0.3% MG and 0% vs. 0.4% MG, respectively). A comparative decrease in body weight gain in MG-treated mice was observed from the first week of administration, but the values became significant from experimental Week 6 to Week 17. However, the health condition of the mice given either dose of MG remained good throughout the study. The average food consumption values were similar in the control and MG-treated groups at 4.5 to 4.8 g/day/mouse. The average MG intake during Weeks 1–17 was 488 mg/kg body weight/day in the 0.3% MG group and 700 mg/kg body weight/day in the 0.4% MG group.

### 2.2. Blood Biochemistry

Blood chemistry data are shown in Figure 2C–H. Control *Apoe*^−/−^ mice showed consistently high levels of total cholesterol (700–800 mg/dL), but mice treated with either 0.3% or 0.4% MG showed significant decreases at Weeks 8 and 17 (Figure 2C,D). Triglyceride levels were similar between the control and 0.3% MG groups (Figure 2E), but significantly decreased in the 0.4% MG group at Weeks 8 and 17 (Figure 2F). The cholesterol lipoprotein profiles were determined at Week 17. The results show that chylomicron remnants and very-low-density lipoprotein (VLDL) levels were significantly lower in mice treated with both 0.3% MG (Figure 2G) and 0.4% MG (Figure 2H) than in the corresponding control mice. However, there were no differences in high-density lipoprotein (HDL) levels between the control and MG-treated mice. We found no statistical differences in the levels of aspartate aminotransferase (AST), alanine aminotransferase (ALT), and blood urea nitrogen (BUN) between the control and MG-treated groups at Weeks 8 and 17 (Appendix A).

### 2.3. Oil Red O Staining of Atherosclerotic Lesions

As shown in Figure 3A,B (0% vs. 0.3% MG and 0% vs. 0.4% MG, respectively), Oil Red O staining showed many atherosclerotic lesions, particularly in the aortic arch, aortic hiatus (diaphragm penetration area), and renal artery bifurcation area. The ratio of Oil Red O-positive staining to tissue area significantly decreased in the aortas of both MG-treated groups compared to the corresponding controls (Figure 3C,D). Additionally, while examining liver tissue, we observed numerous lipid droplets within hepatocytes in the control group (Figure 3E,G). Nevertheless, dietary MG at both concentrations (0.3%, Figure 3F; 0.4%, Figure 3H) markedly decreased hepatocytic lipid droplets in the liver.

### 2.4. Histopathological Analysis

Many atherosclerotic lesions were observed microscopically and classified as early, progressive, and complicated lesions [19]. In early lesions, intimal thickening and foam cells were observed (Figure 4A,B). Progressive lesions (Figure 4C,D) were characterized by a component of foam cell accumulations frequently containing a lipid core with cholesterol crystals covering a fibrous tissue cap. Complicated lesions were progressive and accompanied by calcification or stenosis (Figure 4E,F). Histopathological analysis showed that the average number of atherosclerotic lesions per mouse in both MG-treated groups decreased significantly compared to the corresponding controls (Figure 4G,H). By pathological classification, mice treated with 0.3% MG showed a significant decrease in the average number of early lesions per mouse. Mice given 0.4% MG displayed significant decreases in the average number of early lesions and in progressive lesions per mouse. In addition, while complicated lesions were seen in mice from the control and 0.3% MG groups, no complicated lesions were observed in mice treated with 0.4% MG (Figure 4H).

### 2.5. Biochemical and Molecular Analyses

As shown in Figure 5A,B, the levels of lipoprotein lipase (LPL), a key enzyme in lipid metabolism, were significantly elevated in mice given 0.4% MG versus control mice, but there was no difference in the serum levels of soluble lectin-like oxidized LDL receptor-1 (sLOX-1), a proatherogenic indicator [20]. In hepatic enzymes, although there was no statistical significance in the mRNA levels of *Fatty acid synthase* (*Fasn*) between groups, the levels of *Fatty acid transporter* (*Fatp*; fatty acid uptake of hepatocytes) were significantly suppressed in the MG-treated mice compared to control mice (Figure 5C, top). 3-Hydroxy-3-methylglutaryl-Coenzyme A synthase (HMG-CoA synthase) executes HMG-CoA synthesis from acetoacetyl-CoA. The mRNA levels of *HMG-CoA synthase* were markedly suppressed in the MG-treated mice, whereas the levels of *HMG-CoA reductase*, a cholesterol biosynthetic enzyme from HMG-CoA, were significantly elevated (Figure 5C, bottom). The *HMG-CoA reductase* levels were elevated in the MG-treated mice, possibly due to a compensatory elevation by decreased HMG-CoA associated with *HMG-CoA synthase* inhibition. Real-time PCR analysis of formalin-fixed, paraffin-embedded (FFPE) atherosclerotic lesions revealed no statistical differences in atherogenic markers (*Timp1*, *Mmp9*, *Lox1*, *Cxcl16*, and *Klf5*) between the control and 0.4% MG-treated groups (Figure 5D). We considered that MG exhibited anti-atherosclerosis independent of these atherogenic markers. Next, we analyzed macrophage markers in the lesions.

### 2.6. Quantitative Analysis of Macrophages

Real-time PCR analysis of macrophage indicators showed that expression of *Cd163* (an M2 macrophage marker) was significantly elevated, while levels of *Cd68* (a pan-macrophage marker) and *Nos2* (an M1 macrophage marker) showed no apparent differences between the two groups (Figure 6A). Immunofluorescent double-staining for CD68^+^/NOS2^+^ and CD68^+^/CD163^+^, markers for M1 and M2 macrophages, respectively [21,22], was performed on FFPE atherosclerotic tissues. CD68^+^ staining was characteristic of the whole macrophage population. The numbers of CD68^+^ cells (green cells) tended to be higher in control lesions (Figure 6C,D) than in those exposed to 0.4% MG (Figure 6E,F). In contrast, CD163^+^ macrophages (red cells) were higher in atherosclerotic lesions exposed to 0.4% MG (Figure 6E,F). CD68^+^/CD163^+^ cells (yellow cells), the M2 population, were more frequent in atherosclerotic lesions exposed to 0.4% MG (Figure 6E,F) than in control lesions (Figure 6C,D). It appeared that M1 macrophages (CD68^+^/NOS2^+^) predominated in the atherosclerotic lesions of control mice, whereas the M2 population (CD68^+^/CD163^+^) was elevated in lesions exposed to 0.4% MG, but the increases were variable across individual lesions and not statistically significant (Figure 6B).

### 2.7. M1 and M2 Macrophage-Polarizing Cytokines

As shown in Figure 7, the relative levels of the cytokines *Interferon-γ* (*Ifng*), *Tumor necrosis factor-α* (*Tnfa*), and *Interleukin-1β* (*Il1b*), which induced M1 polarization in the mice [23], tended to be reduced in the atherosclerotic lesions of the 0.4% MG mice, but without statistical significance (Figure 7A). The M2-polarizing cytokines *Interleukin-4* (*Il4*) and *Interleukin-13* (*Il13*), both of which induced M2 polarization in mice, behaved differently. Although the relative levels of *Il4* showed some reduction with 0.4% MG exposure, the levels of *Il13* were significantly elevated in the atherosclerotic lesions (Figure 7B).

## 3. Discussion

Dietary MG significantly suppressed atherosclerotic lesions in the *Apoe*^−/−^ mice and was associated with decreased serum total cholesterol (chylomicron remnants and VLDL) and triglyceride levels, decreased hepatic *HMG-CoA synthase* and *Fatp*, and increased serum LPL activity, in addition to strong attenuation of hepatic steatosis. In addition, real-time PCR showed a significant elevation in *Cd163* (an M2 marker of macrophages), but not *Cd68* (a pan-macrophage marker) and *Nos2* (an M1 marker) of atherosclerotic lesions in mice given MG compared to control mice. Further, immunofluorescence analysis for the classification of macrophages revealed no differences in the M1 population (CD68^+^/NOS2^+^) in atherosclerotic lesions between control and 0.4% MG-treated mice, whereas the M2 population (CD68^+^/CD163^+^) tended to be elevated in the 0.4% MG-treated mice, but no statistical significance was observed because of large variations. In fact, M2-polarizing cytokine *Il13* levels were significantly elevated in the atherosclerotic lesions of mice given 0.4% MG.

Hypercholesterolemia is a well-known high-risk factor for the development of atherosclerosis, and a reduction in blood cholesterol levels leads to a decrease in the risk of atherosclerosis [24]. In the present study, the levels of serum total cholesterol in the control *Apoe*^−/−^ mice were in good agreement with previous reports [19,25], and the administration of both 0.3% and 0.4% MG significantly decreased serum total cholesterol levels and atherosclerosis development compared to control mice. The decreased total cholesterol levels in the MG-treated group may be responsible for the strong suppression of *HMG-CoA synthase* (synthesis of HMG-CoA from acetoacetyl-CoA) and resulted in a decrease of HMG-CoA. Since HMG-CoA reductase conducts cholesterol biosynthesis from HMG-CoA, the levels of *HMG-CoA reductase* were compensatively elevated, possibly due to decreased HMG-CoA by MG administration. In addition, because it is well known that the major atherogenic lipoproteins are LDL and VLDL [26], the cholesterol lipoprotein profiles were characterized and the results showed that both doses of MG induced significant decreases in chylomicron remnants and VLDL. The serum triglyceride levels significantly decreased in the 0.4% MG group. Hypertriglyceridemia has also been reported to be associated with incremental atherosclerosis risk [27]. In addition, it was shown that increased visceral adiposity induces an unfavorable inflammatory cytokine profile such as TNFα [28]. By contrast, adiponectin (a protective adipokine) decreases in obesity [28]. One possible reason for the observed anti-atherogenicity by MG administration might be the reduction in body weight (decreased visceral adiposity).

LPL activity was significantly elevated in the 0.4% MG group. LPL, a key enzyme in lipid metabolism, hydrolyzes triglycerides in circulating triglyceride-rich lipoproteins and facilitates the incorporation of free fatty acids into adipocytes, which are resynthesized into triglycerides and stored [29]. The role of LPL in atherogenesis is controversial. A previous report on a novel compound, NO-1886, notes that it increased LPL activity, resulting in an elevation of HDL cholesterol, and further notes that long-term administration of NO-1886 significantly inhibited experimentally induced atherogenesis in the coronary arteries of rats [30]. Overexpression of LPL protects against atherosclerotic development in human *LPL* transgenic *Apoe*^−/−^ mice [31]. In human *LPL* transgenic low-density lipoprotein receptor-deficient (*Ldlr*^−/−^) mice, overexpression of LPL is associated with decreased triglycerides and triglyceride remnants in the plasma, with a significant reduction of atherosclerotic lesions versus *Ldlr*^−/−^ mice without the human LPL transgene [32]. Human *LPL* transgenic rabbits overexpressing LPL fed a high-cholesterol diet showed hypercholesterolemia but a dramatic reduction of atherosclerotic lesions [33], suggesting that elevated LPL expression protects against atherosclerosis. In contrast, macrophage-derived LPL produced in the arterial wall during the formation of foam cell lesions promoted atherogenesis in *Ldlr*^−/−^ mice [34]. Upregulation of LPL expression induced in *Apoe*^−/−^ mice by miR-182 reportedly promotes lipid accumulation in atherosclerotic lesions and increases proinflammatory cytokine secretion, leading to an acceleration of atherogenesis [35]. Thus, although we still do not know whether LPL acts to enhance or inhibit atherosclerosis, we are certain that it plays a crucial role in atherogenesis. Peroxisome proliferator-activated receptors (PPARs) are also involved in lipid metabolism. They play important roles in the pathogenesis of metabolic disorders, cardiovascular diseases, cancer, and other diseases [36]. γ-MG acts as a dual agonist that activates both PPARα and PPARδ [37], and α- MG increases hepatic PPARγ expression [38]. Since these factors are also considered to be potential therapeutic targets, further investigation of anti-atherogenesis by MG will be needed.

LOX-1 is a major receptor for oxidized LDL in endothelial cells and macrophages [39], and elevated levels of LOX-1 were observed in human atherosclerotic lesions [40]. sLOX-1 is derived from LOX-1 by proteolytic cleavage and released into the blood. In *Apoe*^−/−^ mice, elevated expression of LOX-1 is observed in early atherosclerotic lesions, and expression increases with the progression of disease [19]. Atherosclerosis was shown to be considerably more pronounced in *Lox1* transgenic *Apoe*^−/−^ mice than in *Apoe*^−/−^ mice without the *Lox1* transgene [41]; on the other hand, *Lox1*^−/−^ mice exhibited fewer and less severe atherosclerotic lesions than wild-type mice [42]. We found similar serum sLOX-1 and *Lox1* transcriptional levels in the atherosclerotic lesions of control and MG-treated mice. The expression of other atherogenic biomarkers such as *Timp1*, *Mmp9*, *Cxcl16*, and *Klf5* was also similar between groups, as analyzed by real-time PCR, indicating a different mechanism operating in the MG-induced reduction of atherosclerosis development in *Apoe*^−/−^ mice.

Although hypercholesterolemia is necessary for the development of severe atherosclerotic lesions in *Apoe*^−/−^ mice, *Apoe*^−/−^ mice carrying an *Osteopetrotic* (*Op*) mutation in the macrophage colony-stimulating factor gene show a threefold increase in blood cholesterol levels yet had significantly less atherosclerosis [43], indicating that severe hypercholesterolemia alone is not sufficient for lesion development. Macrophage accumulation within vascular walls is a hallmark of atherosclerosis [23], suggesting that it is an essential function for macrophages during atherogenesis. A shift in macrophage phenotype is widely accepted in plaque biology, in that M2 macrophages shift to M1 macrophages during plaque progression, while M1 macrophages shift to M2 during plaque regression [23,44]. The two phenotypes play different roles in inflammation through the production of proinflammatory cytokines by M1 and anti-inflammatory cytokines by M2. In our study, real-time PCR on samples of atherosclerotic lesions showed a significant elevation in *Cd163* specific for an M2 marker, but not in *Cd68* or *Nos2* (an M1 marker), in mice given MG compared with control samples. Immunofluorescence analysis for macrophage classification revealed no differences in atherosclerotic lesions in the M1 population (CD68^+^/NOS2^+^) between control and MG-treated mice, whereas the M2 population (CD68^+^/CD163^+^) tended to be elevated in MG-treated mice, but not to a statistically significant degree due to large quantitative variations. However, since the M2 population in the lesions of control mice was extremely low with small variations, the increased levels of MG-treated mice were considered to be biologically a meaningful finding. Certain other compounds have been shown to prevent atherosclerosis in *Apoe*^−/−^ mice and be associated with a remarkable shift in M1/M2 macrophage polarization [45]. In addition, M1 macrophage accumulation is a hallmark of the progression of atherosclerosis. Accumulation of cholesterol lipids can direct macrophages toward an M1 proinflammatory phenotype [23]. However, since there were no apparent differences in the numbers of M1 populations between control and MG-treated mice, the anti-atherogenic effect by MG might be strong enough. In fact, treatment with MG did not strongly decrease M1 polarizing cytokines. It may be improved by altering the route of administration or combining with anti-inflammatory drugs.

M1 macrophages are typically induced by specific T_H_1 cytokines (M1 polarization) such as IFN-γ, TNF-α and IL-1β in mice, while M2 macrophages are induced by specific T_H_2 cytokines (M2 polarization) such as IL-4 and IL-13 in mice [23]. Recently, it was reported that α-MG markedly reduced T_H_1 cytokines (IL-6 and TNF-α) in vitro [46]. More recently, MG extract is shown to possess antibacterial activity against *S. aureus* and dramatically downregulate the expression of TH1 cytokines, namely TNF-α, IL-6, and IL-1β, via the TLR-2 pathway in a skin infection model of mice [47]. Furthermore, other studies found that MG decreased the levels of T_H_1 cytokines (TNF-α and IL-1β) [48,49,50]. In the current study, the levels of IFN-γ, TNF-α, and IL-1β (M1 polarizing T_H_1 cytokines) in the atherosclerotic lesions of mice given 0.4% MG tended to decrease, whereas the levels of T_H_2 cytokine *Il13* (IL-13) (an M2 polarizing T_H_2 cytokine) were significantly elevated in the atherosclerotic lesions of mice treated with 0.4% MG as compared with controls, indicating that MG induced the microenvironment for M2 polarization. The plaque regression induced by MG administration may have been due to a switch from the M1 to the M2 phenotype. However, *Il4* levels (IL-4, an M2 polarizing T_H_2 cytokine) showed a tendency to decrease in the 0.4% MG group. One possible reason for this is that, as previously reported, α-MG attenuates MAPK, STAT1, c-Fos, and c-Jun, and subsequently IL-4 production downstream of these cascades is decreased [51].

*Apoe*^−/−^ mice over six months of age developed hepatic steatosis and fibrosis in addition to atherosclerotic lesions [52]. Oil Red O staining of frozen liver sections for lipids revealed numerous lipid droplets within the hepatocytes of the *Apoe*^−/−^ control mice, indicating fatty metamorphosis analogous to simple fatty liver in human nonalcoholic fatty liver disease (NAFLD). In terms of relevance, NAFLD is the most common liver disease in both the adult and pediatric populations [53,54]. Clinically, it is divided into simple fatty liver without inflammation, fibrosis, and steatohepatitis; if left untreated, progression to steatohepatitis is nearly certain. Severe NAFLD due to diabetes mellitus and obesity commonly advances to inflammatory nonalcoholic steatohepatitis (NASH), a precursor of cirrhosis and ultimately hepatocellular carcinoma [55]. Since NAFLD and atherosclerosis share common molecular mediators, NAFLD itself might play a crucial role in the development and progression of atherosclerosis [56]. MG-treated *Apoe*^−/−^ mice, in contrast, exhibited a marked reduction of lipid droplets within the hepatocytes. This may be associated with the decreased levels of total cholesterol (possibly due to *HMG-CoA synthase* inhibition)/triglycerides and *Fatp* and the increased levels of LPL induced by MG noted in the present study. FASN (fatty acid synthase) and FATP (fatty acid transporter) are also involved in lipid metabolism. Hepatic FATP fulfills the function of fatty acid uptake of hepatocytes and FASN is a key enzyme required for de novo synthesis of fatty acid, and the lipid homeostasis needs a fine-tuning of *Fatp* [57]. Although α-MG is a potent inhibitor of FASN in vitro study [58], the present in vivo study did not show inhibition of *Fasn* transcriptional levels. Since the primary function of *Apoe* is the hepatic and extra-hepatic uptake of plasma lipoprotein and cholesterol [59], it may be one of the reasons for alterations in lipid metabolism in *Apoe* null mice. Aberrations in lipid metabolism by *Fatp* also elicit a pathological state (lifestyle-related disease) [57]. The observed attenuation of hypercholesterolemia and hepatic steatosis in MG-treated mice in the present study may be due to the suppression of *Fatp* and *HMG-CoA synthase*. In a recent study, it is shown that α-MG reduced fat mass accumulation and lipid profiles such as cholesterol, triglycerides, and fatty acid in obese mice [38]. Furthermore, α-MG decreased hepatic steatosis through the hepatic SirT1-AMPK and PPAPγ pathways in obese mice [38]. In addition, it is reported that α-MG ameliorated adipose inflammation and hepatic steatosis in obese C57BL/6 mice receiving a high-fat diet [60], corroborating our findings in this study.

## 4. Materials and Methods

### 4.1. MG Extraction and Purification

MG comprising 84% α-MG (Figure 1C) and 7% γ-MG (Figure 1D) was obtained from Ecoresource Institute Co. Ltd. (Gifu, Japan). Briefly, mangosteen (*Garcinia mangostana* Linn) pericarps collected in Thailand were dried, ground, and immersed in hot water, and the filtered residue was extracted in 50% ethanol. The filtrate was concentrated and vacuum-dried. The resultant dried substance was dissolved in ethanol, gradually added to water, and crystallized. The crystallized powder was repeatedly resuspended in ethanol and water, and recrystallized until it reached the prescribed purity range of 75–85% α-MG and 5–15% γ-MG.

### 4.2. Animals

Homozygous mouse mutants for *Apolipoprotein e* (*Apoe^tm1Unc^*; *Apoe*^−/−^) on a C57BL/6J background were obtained from Jackson Laboratory (Bar Harbor, ME, USA). Homozygous mutation of the *Apoe* results in functional knockout of the anti-atherogenic *Apoe* gene involved in cholesterol metabolism [59]. A total of 45 *Apoe*^−/−^ mice were used in this study. Mice were bred and maintained in the animal facility of the National Cerebral and Cardiovascular Center Research Institute (Osaka, Japan), and housed at a density of 5 mice per plastic cage on woodchip bedding with free access to water and food under controlled temperature (22 ± 3 °C), humidity (50 ± 10%), and lighting (12 h/12 h light/dark cycle).

All animal experiments were approved by the Institutional Review Board of the National Cerebral and Cardiovascular Center Research Institute, Japan (approval numbers 13047, 14007, 15038, 16032, and 17072; 22 March 2017) and were performed in accordance with the procedures outlined in the Guide for the Animal Care Ethics Committee of the institute.

### 4.3. Experimental Design

Since it was difficult to obtain the required number of *Apoe*^−/−^ mice due to a low reproductive rate, the study was performed in 3 separate experiments. In Experiment 1, 16 7-week-old male *Apoe*^−/−^ mice were randomly divided into 2 groups of 8 mice each. One group was fed a basal diet as control (CE-2, Clea Japan Inc., Tokyo, Japan), and the other group received 0.3% MG mixed with the basal diet for 17 weeks. In Experiment 2, 12 7-week-old male *Apoe*^−/−^ mice were randomly divided into 2 groups of 6 mice each; one group received the basal control diet, and other received the basal diet with 0.4% MG. Food consumption was measured over a 2-day period in each week per cage: food intake values divided by number of surviving mice (in a cage) multiplied by 2 days = food consumption per mouse per day (within a week). Then, average food consumption values during the experimental period were calculated from food consumption values in each week. Both experiments extended for 17 weeks. At 8 and 17 weeks, surviving mice were deprived of food overnight, and blood was collected from tail veins for blood lipid measurements. Dietary dosages of MG were determined based on the results of a previous study (0, 0.25, and 0.5%) [7].

Experiment 3 included 17 7-week-old male *Apoe*^−/−^ mice providing tissue solely for elucidation of the mechanisms by real-time PCR and quantitative immunohistochemical analysis for macrophages as the process of Oil Red O stain-induced RNA degradation. In this part of the study, 8 mice received the basal control diet and 9 mice received 0.4% MG in the same basal diet for 17 weeks. At the termination of the study at 17 weeks, the animals were euthanized and necropsied, and all organs were fixed in 10% phosphate-buffered formalin overnight. In addition, a portion of the livers was immediately frozen in liquid nitrogen for molecular analysis (liver enzymes).

### 4.4. Blood Chemistry and Oil Red O Staining (Experiments 1 and 2)

At experimental Weeks 8 and 17, whole-blood samples were drawn from the tail veins of mice from experiments 1 and 2 using blood collection tubes for infants (BD Microtainer, Becton and Dickinson Co., Franklin Lakes, NJ, USA), and centrifuged at 6000× *g* for 2 min. Samples were then quantified for total cholesterol, triglyceride, AST, ALT, and BUN levels using a Fuji Dry-Chem analyzer (7000 V; Fuji Film Co., Tokyo, Japan).

After the excision of abdominal organs and lungs at study termination, hearts with the entire ascending to descending aorta from the aortic root to the bifurcation of the common iliac artery were carefully separated from the mice. After overnight fixation, extravascular adipose tissues were removed, and the aortae were stained with Oil Red O as previously described [19]. Briefly, the aortae were opened lengthwise, fastened flat with pins on a wax dissection board, and stained with 0.3% Oil Red O (FUJIFILM Wako Pure Chemical Co., Osaka, Japan) in 2-propanol and distilled water (6:4) at 37 °C for 15 min. The tissues were further rinsed with 2-propanol for 1 min and dipped in distilled water, and the distribution of the atherosclerotic lesions was recorded. Oil Red O-positive areas, presumed to be atherosclerotic lesions, and total area of whole aorta were measured with the WinROOF software program (Mitani Corp., Tokyo, Japan) and expressed as percentage of the whole aorta.

In addition, samples of livers were embedded in Tissue-Tek OCT compound (Sakura Finetek Inc., CA, USA), frozen in liquid nitrogen, and stored at −80 °C until use. Frozen livers were sliced into 5 μm thickness using a cryostat (Leica Biosystems GmbH, Nussloch, Germany), slide-mounted, and dried. Once fully dried, liver slides were immersed in distilled water and stained with 0.3% Oil Red O with hematoxylin counterstaining.

### 4.5. Histopathological Evaluation of Atherosclerotic Lesions (Experiments 1 and 2)

Aortae were trimmed at 2–3 mm from the aortic root to the aortic arch and at 4–5 mm through the descending thoracic and abdominal aorta. Ascending aortic slices were horizontally embedded in paraffin, while the sections of descending aorta were embedded longitudinally. FFPE tissues were sectioned into 3–4 μm thickness, mounted, and stained with hematoxylin/eosin (H&E) or Elastica van Gieson stains.

Atherosclerotic lesions were histologically classified into 3 categories [19]: early, progressive or advanced, and complicated. Simple intimal thickening and foam cell accumulation are characteristic of early lesions. Progressive/advanced lesions are marked by inflammatory cell infiltrations, including foam cells and leukocytes, and complicated lesions are accompanied by secondary lesions such as calcification or stenosis.

### 4.6. Cholesterol Lipoprotein Profiles, Blood Chemistry for LPL and sLOX-1, and Real-Time PCR Analysis of FFPE Tissues (Experiment 3)

Cholesterol lipoprotein profiles of the serum from control and 0.4% MG-treated *Apoe*^−/−^ mice were determined by gel-filtration high-performance liquid chromatography at Skylight Biotech Inc. (Akita, Japan). High levels of LPL activity are reported to have a protective effect against atherosclerosis development [31,32,33], while serum sLOX-1 is associated with atherosclerosis development [40,61]. For this study, we measured the LPL activity in serum from mice receiving 0.4% MG and control mice using the LPL Activity Assay kit (Roar Biomedical Inc., New York, NY, USA). The levels of serum sLOX-1 were determined using a commercial enzyme-linked immunosorbent assay (Abcam, Cambridge, MA, USA).

We also measured the mRNA levels of the liver enzymes using real-time PCR. Total RNA was isolated from livers using a RNeasy Mini Kit (Qiagen GmbH, Hilden, Germany). We used a High-Capacity cDNA Reverse Transcription Kit (Applied Biosystems, Foster CA, USA) to prepare cDNA from 10 µg of each sample of the total RNA. Each cDNA sample was subjected to real-time PCR (StepOnePlus Real-Time PCR System; Applied Biosystems) to measure the relative quantities of *Glyceraldehyde-3-phosphate dehydrogenase* (*Gapdh*), *Fasn*, *Fatp2*, *HMG-CoA synthase* (*Hmgcs2*) and *HMG-CoA reductase* (*Hmgcr*) expression using SYBR Green reagent (Applied Biosystems) with the TaqMan Gene Expression Assay (Applied Biosystems). The expression levels of target genes were normalized using *Gapdh* expression as an internal control. TaqMan Gene Expression Assay; assay ID for *Fasn*, *Fatp2*, *HMG-CoA synthase* and *HMG-CoA reductase* were Mm00662319, Mm0049877, Mm00550050 and Mm01282492, respectively. After an amplification consisting of 40 cycles, the relative expression of each target was analyzed with software of the Applied Biosystems.

In addition, we analyzed FFPE aorta samples from mice that received 0.4% MG and control mice using real-time PCR, looking for relative levels of atherogenic factors (*Timp1*, *Mmp9*, *Lox1*, *Cxcl16*, and *Klf5*), macrophage markers (*Cd68*, *iNos2*, and *Cd163*), and macrophage-polarization cytokines (*Ifng, Tnfa*, *Il1b*, *Il4* and *Il13*). Total RNA was isolated from FFPE tissues using the RNeasy FFPE kit (Qiagen GmbH), and genomic DNA elimination was conducted. cDNA was then synthesized using a Primer Script RT Reagent kit with gDNA Eraser (Takara Bio, Otsu, Shiga, Japan), according to the manufacturer’s instructions. Amplification of the obtained cDNA was performed using a Thermal Cycler Dice Real-Time System Single (model TP850, Takara Bio) with SYBR Premix Ex Taq II (Tli RNase H Plus, Takara Bio), according to the manufacturer’s instructions. *Gapdh* was used as an internal control. The PCR conditions were as follows: an initial step at 95 °C for 30 s, followed by 45 cycles of 95 °C for 5 s, 58 °C for 10 s, and 72 °C for 20 s. The quality of total RNA extracted from the FFPE samples was confirmed through *Gapdh* amplification dissociation curves. Samples that showed poor *Gapdh* amplification, or amplification or aberrant dissociation temperature of amplified products, were excluded from later analysis. The primer sequences used for the mouse FFPE samples are shown in Table 1. The relative expression of the amplified products was normalized to *Gapdh* (ΔCt) and expressed as fold change calculated using the 2^−ΔΔCt^ method as described previously [62].

### 4.7. Quantitative Immunofluorescence Analysis (Experiment 3)

We employed immunofluorescence staining to analyze the macrophage markers CD68, NOS2, and CD163 in the atherosclerotic lesions of the control and 0.4% MG mice, using the labeled streptavidin-biotin (LSAB) method (Dako, Glostrup, Denmark). Because both M1 and M2 phenotypes share the expression of CD68, we performed immunofluorescence double-staining for CD68^+^/NOS2^+^ as markers for M1 macrophages and CD68^+^/CD163^+^ as markers for M2 macrophages [22]. The primary antibodies used in this study were as follows: anti-CD68 rat monoclonal antibody (Novus Biologicals, Littleton, CO, USA), anti-iNOS rabbit polyclonal antibody (Thermo Scientific, Fremont, CA, USA), and anti-CD163 rabbit polyclonal antibody (Bioss, Woburn, MA, USA). Unstained sections were immersed in Tris-EDTA buffer (pH 9.0) and antigen retrieval was performed at 110 °C for 10 min using an autoclave prior to incubation with the primary antibodies. The slides were exposed to the secondary antibodies anti-rat Alexa Fluor 488 (for CD68) and anti-rabbit Alexa Fluor 594 (for NOS2 and CD163) (Molecular Probes, Eugene, OR, USA) with nuclear staining achieved by mounting in medium containing 4’,6-diamidino-2-phenylindole (DAPI) (Vector Laboratories, Burlingame, CA, USA). Prior to the quantitative analysis, autofluorescence normalization was conducted on unstained sections by a Mantra quantitative pathology workstation with inForm Image Analysis software (PerkinElmer, Boston, MA, USA). We were then able to perform quantitative analysis on the numbers of cells expressing CD68, iNOS, and CD163.

### 4.8. Statistical Analysis

When the data were normally distributed, an unpaired-group Student’s *t*-test was used to compare the values between the control and MG-treated groups. When there was insufficient homogeneity of variance, we used Welch’s method. The Mann–Whitney U-test was used in cases of non-normal data distribution. Differences were considered statistically significant at *p* < 0.05.

## 5. Conclusions

MG, composed largely of α-mangostin, suppressed the development of atherosclerosis and associated changes in the serum lipid biochemistry of *Apoe*^−/−^ mice, possibly through an M2 macrophage-mediated mechanism. A schematic presentation of MG-induced anti-atherosclerosis is shown in Figure 8. In addition, MG decreased the severity of the fatty liver characteristics of *Apoe*^−/−^ mice. MG may thus provide beneficial ameliorating effects on the development of atherosclerosis caused by lifestyle.

## Figures and Tables

**Figure 1 ijms-20-01722-f001:**
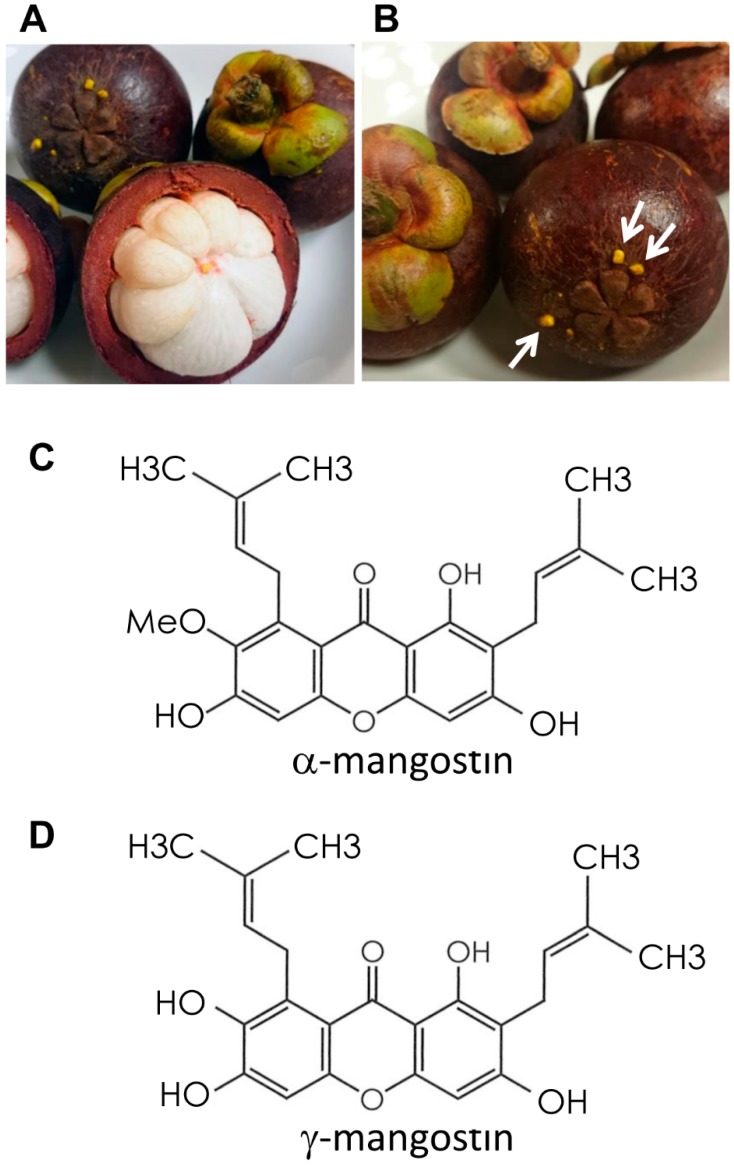
Gross appearance of mangosteen and chemical structures of α- and γ-mangostin (MG). (**A**) Mangosteen is a round fruit with a thick, deep purple spherical outer pericarp. The edible snow-white endocarp is composed of four- to eight-segmented wedge-shaped arils. (**B**) When damaged, the mangosteen pericarp secrets a yellow exudate for protection from infection and insects. Chemical structure of mangosteen extract: (**C**) α-MG, molecular formula C_24_H_26_O_6_ and molecular weight 410; and (**D**) γ-MG, molecular formula C_23_H_24_O_6_ and molecular weight 396.

**Figure 2 ijms-20-01722-f002:**
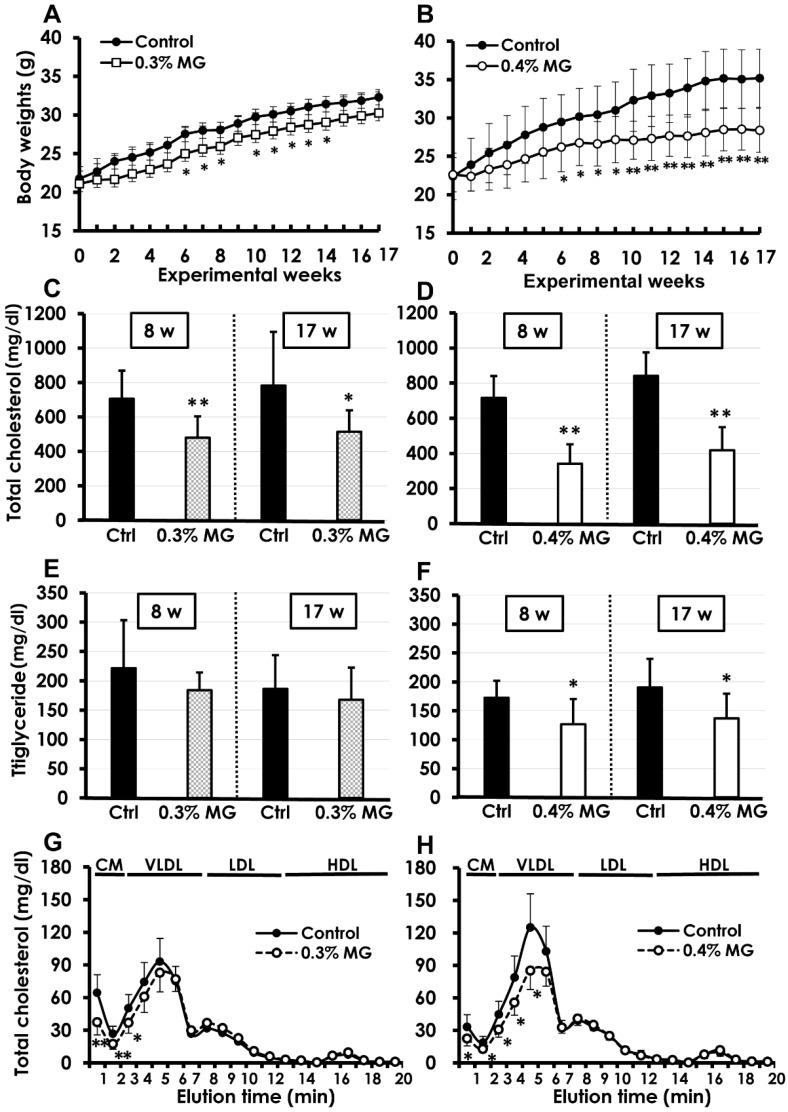
Body weights and blood biochemistry of *Apoe*^−/−^ mice administered crude dietary α-MG at (**A**) 0.3% and (**B**) 0.4%. Body weights of mice given either dose of MG tended to be lower than those of controls throughout the study; differences became significant at Week 6 and continued until the end of the study at Week 17. Total serum cholesterol levels significantly decreased over controls in both the (**C**) 0.3% and (**D**) 0.4% MG groups sampled at Weeks 8 and 17. Serum triglyceride levels were similar between control and (**E**) 0.3% MG-treated mice but significantly decreased in (**F**) 0.4% MG-treated mice sampled at Weeks 8 and 17. Cholesterol lipoprotein profiling at Week 17 showed that chylomicron remnants and very-low-density lipoprotein (VLDL) levels were significantly lower in both (**G**) 0.3% and (**H**) 0.4% MG treated mice than in corresponding control mice. In contrast, there were no differences in high-density lipoprotein (HDL) levels between control and MG-treated mice. * *p* < 0.05, ** *p* < 0.01 compared to control group. Data are presented as mean ± standard deviation (SD). Eight mice were examined in the control and 0.3% MG groups at Weeks 8 and 17, and six and five in the control and 0.4% MG groups at Weeks 8 and 17, respectively.

**Figure 3 ijms-20-01722-f003:**
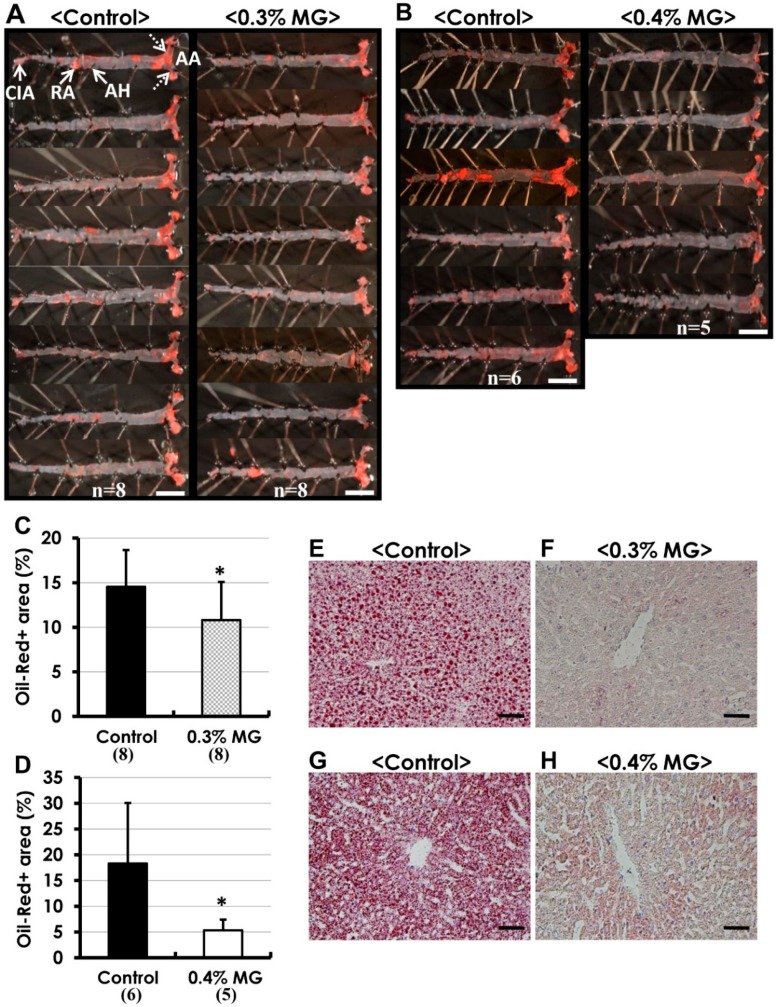
(**A**,**B**) Oil Red O staining of atherosclerotic lesions in *Apoe*^−/−^ mice administered MG in diet; macroscopically, atherosclerotic lesions appear as reddish areas on the aortic lumen. There were fewer lesions in mice exposed to (**A**) 0.3% and (**B**) 0.4% MG compared to corresponding control mice. Percentage of Oil Red O-positive tissues significantly decreased with exposure to (**C**) 0.3% and (**D**) 0.4% MG compared to the corresponding control groups. (**E**–**H**) Dietary MG attenuated hepatic steatosis in *Apoe*^−/−^ mice. Frozen liver sections stained with Oil Red O for visualization of lipids revealed that (**E**,**G**) control *Apoe*^−/−^ livers contained numerous lipid droplets within hepatocytes, indicating fatty metamorphosis, whereas lipid droplets were dramatically reduced in hepatocytes exposed to either (**F**) 0.3% or (**H**) 0.4% MG. (**A**,**B**,**E**–**H**) Oil Red O stain. Scale bars: (**A**,**B**) 0.5 cm; (**E**–**H**) 50 μm. * *p* < 0.05 compared to control. Data are presented as mean ± SD. Numerals in parentheses indicate numbers of animals examined. AA, aortic arch (broken arrows); AH, aortic hiatus; RA, renal artery; CIA, common iliac artery.

**Figure 4 ijms-20-01722-f004:**
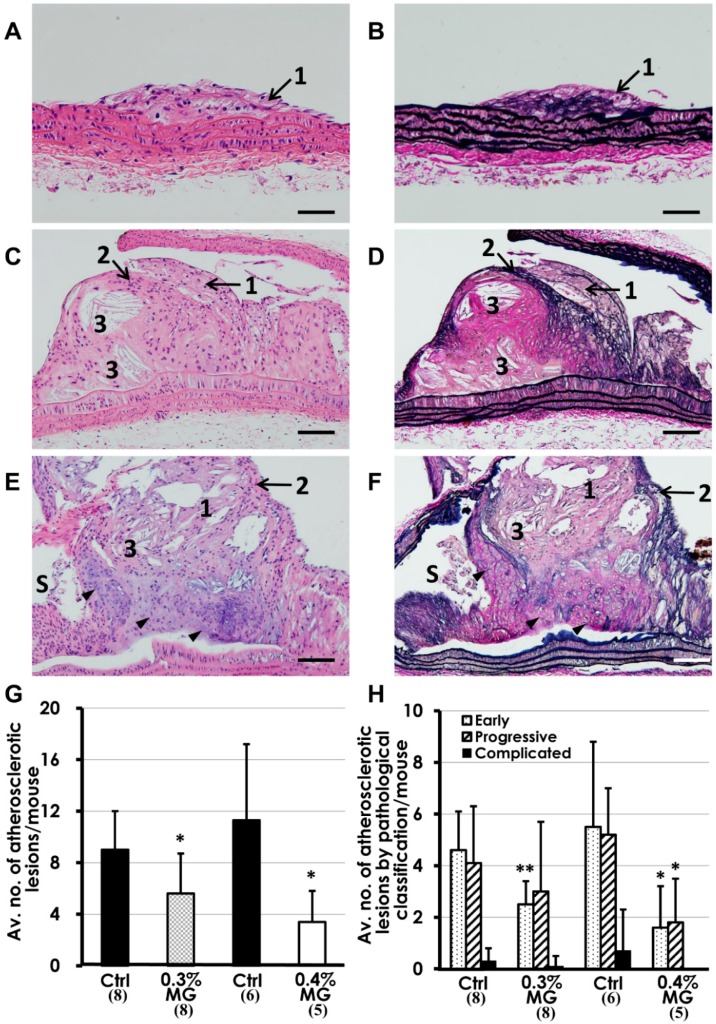
Histopathology of atherosclerotic lesions in *Apoe*^−/−^ mice administered dietary MG. Lesions are classification as (**A**,**B**) early, with simple foam cell accumulations (indicated by 1); (**C**,**D**) progressive, with development of a fibrous cap (indicated by 2) and cholesterol crystals within lipid cores (indicated by 3) in addition to foam cells (1); and (**E**,**F**) complicated, containing all of the above characteristics but with stenosis (indicated by S) and areas of ossification or chondrification (arrowheads). (**G**) Quantitatively, the average number of atherosclerotic lesions per mouse significantly decreased in both 0.3% and 0.4% MG groups compared to corresponding control groups. (**H**) In mice given 0.3% MG, the number of early lesions per mouse significantly decreased, and the number of early and progressive lesions per mouse significantly decreased with 0.4% MG exposure compared to the corresponding controls. (**A**–**F**) Atherosclerotic lesions from control *Apoe*^−/−^ mice administered basal diet; (**A**,**C**,**E**) H&E stain; (**B**,**D**,**F**) Elastica van Gieson. Scale bar: (**A**,**B**) 50 μm; (**C**–**F**) = 100 μm. (**G**,**H**) * *p* < 0.05, ** *p* < 0.01 versus control. Data are presented as mean ± SD. Numerals in parentheses indicate numbers of animals examined.

**Figure 5 ijms-20-01722-f005:**
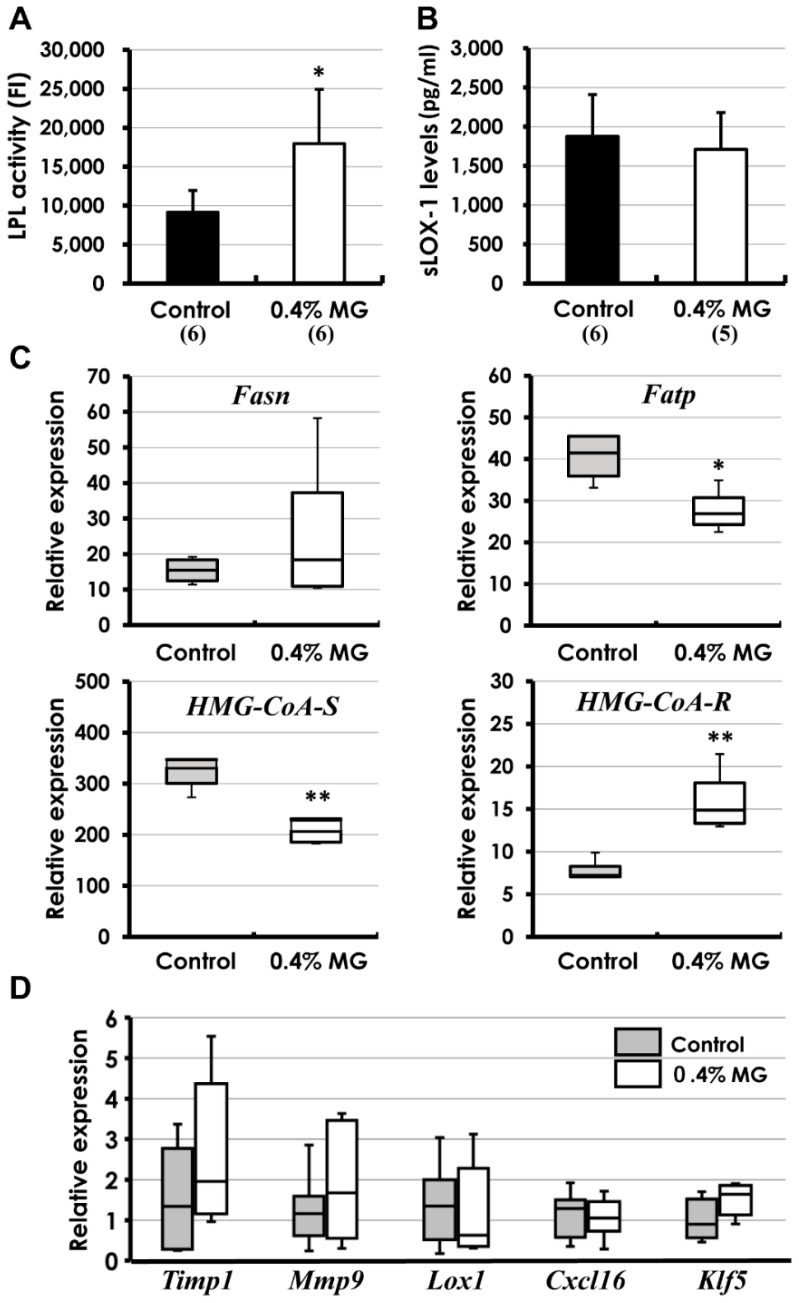
Mechanism analysis using blood biochemistry and real-time PCR. (**A**) Serum lipoprotein lipase (LPL) activity was significantly elevated in *Apoe*^−/−^ mice administered 0.4% MG versus control group. (**B**) Although moderately high (normal level was approximately 1000 pg/mL), serum soluble lectin-like oxidized LDL receptor-1 (sLOX-1) levels were similar in the control and 0.4% MG-treated mice. (**C**) There was no statistical difference in the mRNA levels of *Fasn* between the control and MG-treated mice. However, *Fatp* expression was significantly inhibited in MG-treated mice compared to control mice. *HMG-CoA synthase* (*HMG-CoA-S*) levels were very suppressed in MG-treated mice, whereas *HMG-CoA reductase* (*HMG-CoA-R*) levels were significantly elevated. The elevation in *HMG-CoA-R* levels may be due to a compensatory alteration by decreased HMG-CoA associated with *HMG-CoA-S* inhibition by MG administration. (**D**) Similarly, real-time PCR analysis of formalin-fixed paraffin-embedded (FFPE) lesions for atherogenic markers (*Timp1*, *Mmp9*, *Lox1*, *Cxcl16*, and *Klf5*) revealed no apparent differences between control and 0.4% MG. (**A**–**C**): * *p* < 0.05; ** *p* < 0.01 compared to corresponding control group. Data are presented as mean ± SD. Numerals in parentheses indicate numbers of animals examined. In (**C**,**D**), boxes represent 25th to 75th percentiles, and horizontal lines within boxes represent median values. The whiskers extend to the 10th and 90th percentiles. Eight mice were examined from the control group, and nine mice from the 0.4% MG group, with the exception of *Mmp9* values (*n* = 7).

**Figure 6 ijms-20-01722-f006:**
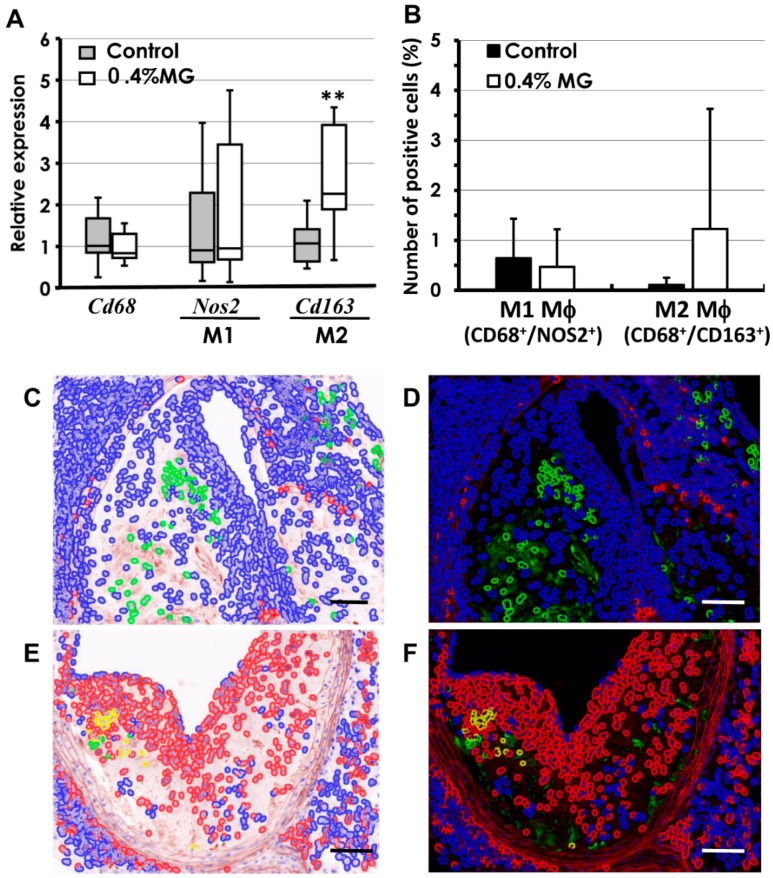
Quantitative analysis of macrophages was conducted. (**A**) Real-time PCR analysis showed that the level of *Cd163* expression, an indicator of M2 macrophage activity, was significantly elevated in mice given 0.4% MG, whereas the levels of *Cd68* (M1 and M2) and *Nos2* (M1) were not markedly changed. (**B**–**F**) Identification and quantification of macrophage subtypes in atherosclerotic lesions by immunofluorescent staining. Number of CD68^+^ cells (green) tended to be higher in (**C**,**D**) control lesions than (**E**,**F**) those exposed to 0.4% MG. Conversely, CD163^+^ cells (red) tended to be more numerous in (**E**,**F**) lesions exposed to 0.4% MG than in (**C,D**) control lesions. Since the expression of CD68 is shared by M1 and M2 phenotypes, immunofluorescent double staining for CD68^+^/NOS2^+^ and CD68^+^/CD163^+^ was used as a marker for M1 and M2, respectively. CD68^+^/CD163^+^ cells (yellow) were counted as M2 macrophages and were more frequent in the lesions of mice given 0.4% MG (yellow cells in (**E**,**F**)), whereas no CD68^+^/CD163^+^ cells (yellow) were seen in control lesions (**C**,**D**). (**B**) Quantification of double-positive cells revealed similar numbers of M1 cells in control and 0.4% MG lesions, whereas M2 populations tended to have more numerous lesions in 0.4% MG-treated mice; however, the differences were not statistically significant because of large variations. (**A**) ** *p* < 0.01 compared to corresponding control group. Boxes represent 25th to 75th percentiles, and horizontal lines within boxes represent median values. The whiskers extend to the 10th and 90th percentiles. (**B**) Macrophages; data are presented as mean ± SD; *n* = 7 in each group. (**C**–**F**) Immunofluorescent staining for CD68 (green) using Alexa Fluor 488 and CD163 (red) using Alexa Fluor 594. (**C**,**E**) pseudo-H&E images of CD68 and CD163 with 4’,6-diamidino-2-phenylindole (DAPI); (**D**,**F**) dark field. Scale bar: (**C**–**F**) 50 μm.

**Figure 7 ijms-20-01722-f007:**
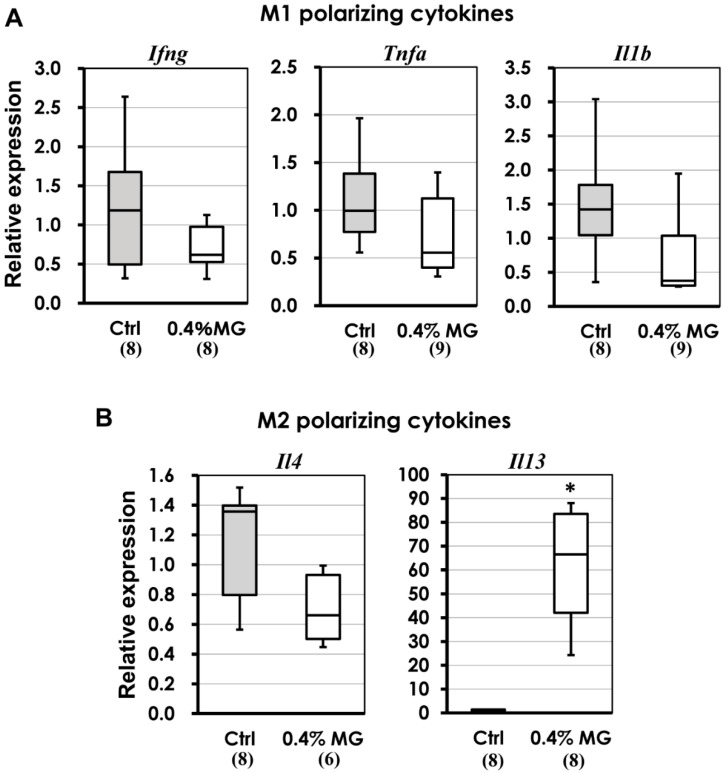
Expression of polarizing cytokines specific to M1 and M2 macrophages associated with atherosclerotic lesions as demonstrated by real-time PCR analysis of FFPE tissues. (**A**) In M1 polarizing cytokines, the relative levels of *Ifng* (*Interferon-γ*), *Tnfa* (*Tumor necrosis factor-α*) and *Il1b* (*Interleukin-1β*) tended to be reduced with 0.4% MG administration compared to control levels, but not significantly. (**B**) In M2 polarizing cytokines, *Il13* (*Interleukin-13*) levels were significantly elevated over control, while *Il4* (*Interleukin-4*) expression levels tended to be reduced with 0.4% MG exposure. * *p* < 0.05 compared to corresponding control tissues. Numerals in parentheses indicate numbers of animals examined. In the case of *Il13*, five mice from the control group and three mice from the 0.4% MG group showed levels under the detection limits. Boxes represent 25th to 75th percentiles, and horizontal lines within boxes represent median values. The whiskers extend to the 10th and 90th percentiles. Numerals in parentheses indicate numbers of FFPE atherosclerotic samples examined per group.

**Figure 8 ijms-20-01722-f008:**
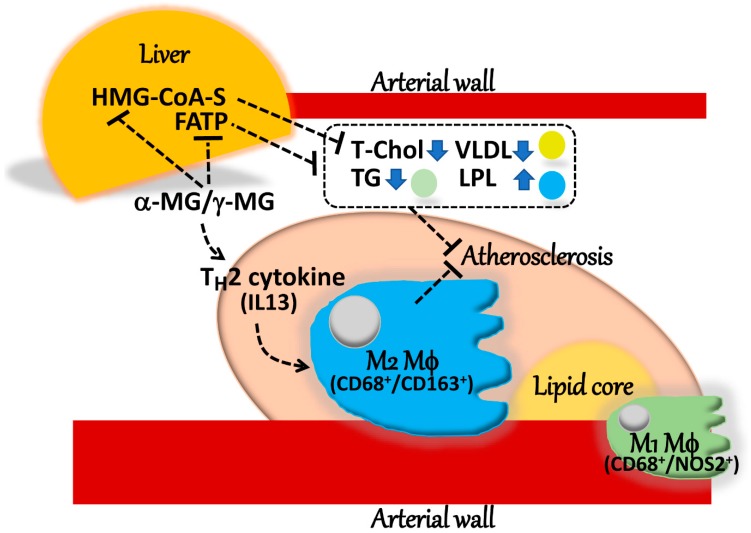
Schematic presentation of MG-induced anti-atherosclerosis. φ, macrophage; MG, mangostin; IL, interleukin; HMG-CoA-S, HMG-CoA synthase; FATP, fatty acid transporter; T-Chol, total cholesterol; TG, triglycerides; LPL, lipoprotein lipase. Break lines ending in arrowheads denote activation and break lines ending in bars represent inhibition.

**Table 1 ijms-20-01722-t001:** Mouse primer sequences for real-time PCR.

Gene	Forward Primer Sequence (5′-->3′)	Reverse Primer Sequence (5′-->3′)
*Timp1*	CAGAACCGCAGTGAAGAGT	CAAGGGATAGATAAACAGGGAAAC
*Mmp9*	ATTCGCGTGGATAAGGAGTTC	GGCAGAAATAGGCTTTGTCTTG
*Lox1*	CTCTGGCATAAAGAAAACTGTTACC	GCTTCTTCCGATGCAATCC
*Cxcl16*	CCGCAGGGTACTTTGGATCAC	GGACTGCAACTGGAACCTGA
*Klf5*	CTGTCAGATACAACAGAAGGAGT	GTGAGCTTTTAAGTGAGACGAC
*Cd68*	GTCTCTCTCATTTCCTTATGGACAG	GCTCTGATGTAGGTCCTGTTTG
*Nos2*	GGGCAGTGGAGAGATTTTG	TGCAAGTGAAATCCGATGTG
*Cd163*	AGTCCTGGATCATCTGTGACAAC	ACACGTCCAGAACAGTCTGTATG
*Infg*	CATGAGTATTGCCAAGTTTGAGGTC	ATTGAATGCTTGGCGCTGGA
*Tnfa*	CACGCTCTTCTGTCTACTGAACT	GAGGCCATTTGGGAACTTCTCAT
*Il1b*	GAGAATGACCTGTTCTTTGAAGTTG	GTTTGGAAGCAGCCCTTCAT
*Il4*	GATGCCTGGATTCATCGATAAG	GTCTTTCAGTGATGTGGACTTG
*Il13*	GCTGAGCAACATCACACAAG	CATACCATGCTGCCGTTG
*Gapdh*	TGGCCTTCCGTGTTCCTACC	AGCCCAAGATGCCCTTCAGT

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
