# Peer review of "Crude α-Mangostin Suppresses the Development of Atherosclerotic Lesions in Apoe-Deficient Mice by a Possible M2 Macrophage-Mediated Mechanism"

_ijms, 2019, doi:10.3390/ijms20071722_

Round 1
Reviewer 1 Report
The manuscript by Shibata et al investigates the effect of crude α-Mangostin on the development of atherosclerotic lesions in apoE-deficient mice. The authors identified changes in biochemical and histopathology upon MG treatment and provide evidence to support that suppression of atherosclerosis is possible mediated through an M2-macrophage-mediated mechanism. This is an interesting finding, however, additional experimental evidence would be required to support their findings. In addition, clarifications are required at various points in order to explain the rationale behind experimental design.
In specific,
-What is the rationale behind the MG mixture chosen to be used in this study (84% α- and 7% γ-MG)? Why were these percentages of a- and g-MG chosen? Please explain. Also, please explain why 0.3% and 0.4% MG concentrations were chosen for treatment of mice? Are these concentrations comparable to the ones previously reported?
- In Figure 2B, the control mice in panel B exhibit great variability in body weight, as shown by their large SD values, but this is not observed in control mice of panel A. How can this happen?
-Why were timepoints for blood biochemistry data analysis chosen to be at 8 and 17 weeks?
-In Figure 4, at the histopathology panels A-F it is not clear what mice they refer to, control, 0.3% or 0.4% MG treated mice? Please clarify.
-Fig 6 legend explains that panels C and E are light field microscopy images. Given that fluorescent cells are also shown at these panels, this should be changed to either overlay of light and dark microscopy fields or the authors should show only the light field image. Currently these two panels are confusing and misleading.
-In section 4.2 of the methods, it is mentioned that 5 C57BL/6 animals were used. For which studies were these animals used? My understanding on the design and findings presented in this study is that animals used were control (untreated) apoE-/- and MG-treated apoE-/- mice. Is this not correct?
-While most studies were performed in 0.3% and 0.4% treatments, in section 2.5 biochemical and molecular analysis were performed only with 0.4% MG. Why?
-Given that biochemical and histopathological changes were detected by MG treatment, how do the authors explain that they did not observe any differences in atherogenic markers?
-Given the effect of MG on body weight and lipid metabolism, and since adipose tissue releases inflammatory factors, could the observed anti-inflammatory effects be due to the reduced body weight and fat of MG-treated mice?
- While the effect of MG on TH1 cytokines has been well documented, as discussed on page 14, how do the authors explain that in their study no significant effect was seen on IFNg, Tnfa and Il1b inflammatory genes?
-Since from the analysis of the 2 anti-inflammatory cytokines (Il-4 and Il-13) only Il-13 showed to be altered by MG treatment, do the authors have any evidence on the effect of MG on additional anti-inflammatory markers (eg NFK-b)?
- Given the controversial role of LPL on atherogenesis, do the authors have any evidence on additional lipid metabolism enzymes (eg. PPARg, fatty acid synthase) in MG-treated mice?
-Could the authors provide a scheme/figure of their overall hypothesis regarding the underlying atheroprotective mechanism of MG based on their finding on the effect of MG on apoE-/- mice.
Author Response
Responses to Reviewer 1
1. Comment #1: What is the rationale behind the MG mixture chosen to be used in this study (84% α- and 7% γ-MG)? Why were these percentages of a- and g-MG chosen? Please explain. Also, please explain why 0.3% and 0.4% MG concentrations were chosen for treatment of mice? Are these concentrations comparable to the ones previously reported?
Responses: (a) We conducted a preliminary study using Apoe-/- mice given 0.3% or 0.4% pure a-MG (not crude MG) with food. The results showed significant inhibition of the weight gain of mice given 0.4% a-MG, and a remarkable reduction in body weight was observed in some animals. Additionally, dietary 0.3% pure a-MG also caused a significant reduction in weight gain (much milder than that of the 0.4% group) with a good general health condition. Because we chose crude MG (containing 80% a-MG) not pure a-MG, in the present study, 0.3% and 0.4% of crude MG in the diet were assumed to be safe. In addition, pure a-MG is extremely expensive; it costs $40,000 for a mouse feeding study. However, the cost of MG extract (mixture of 75-85% a- MG and 5-15% g- MG) is $10,000. Another reason why the MG mixture was used in the present study was economics.
(b) a- and g- MG are contained in mangosteen pericarp at ratio of 75-85% a- MG and 5-15% g- MG. The MG extract used in the present study was 84% α- and 7% γ- MG.
(c) We previously conducted an MG extract feeding study (0.5, 0.25, 0% in diet) in mice (wild-type BALB/c) (Anticancer Res., 29, 2485-2496, 2009). In the present study, since Apoe-/- mice were weak when compared to wild-type mice, we chose 0.4% MG extract as a high dosage (maximum tolerable dose). We chose 0.3% as a second dosage, because we also wanted to observe the therapeutic effect for atherosclerosis. The description regarding the determination of dietary dosages is in Section 4.3 Experimental design: – “Dietary dosages of MG were determined based on the results of a previous study (0, 0.25 and 0.5%)(Anticancer Res, 29, 2485, 2009) and a preliminary study (0 and 0.4%) (unpublished data).” Additionally, we have added the measurement method of food consumption values to this section (Section 4.3 Experimental design).
2. Comment #2: In Figure 2B, the control mice in panel B exhibit great variability in body weight, as shown by their large SD values, but this is not observed in control mice of panel A. How can this happen?
Responses: Since it was difficult to obtain the required number of Apoe−/− mice due to a low reproductive rate, the study was performed in separate experiments. Although all Apoe-/- mice were genetically the same, there were somewhat different by litter; for example, in the incidence of malocclusion. Therefore, the large SD values in Figure B may be due to different litters.
3. Comment #3: Why were timepoints for blood biochemistry data analysis chosen to be at 8 and 17 weeks?
Responses: Since the period of the study was 17 weeks, time points for blood collection were determined at an interim point (8 weeks) and a terminal point (17 weeks).
4. Comment #4: In Figure 4, at the histopathology panels A-F it is not clear what mice they refer to, control, 0.3% or 0.4% MG treated mice? Please clarify.
Responses: Photograph-derived groups are described in the legend of Figure 4.
5. Comment #5: Fig 6 legend explains that panels C and E are light field microscopy images. Given that fluorescent cells are also shown at these panels, this should be changed to either overlay of light and dark microscopy fields or the authors should show only the light field image. Currently these two panels are confusing and misleading.
Responses: Slides were imaged at 100x using imaging software on a Mantra Quantitative Pathology Work station (PerkinElmer). Photographs C and E in Figure 6 are pseudo-H&E images of CD68 and CD163 with DAPI. The description has been added in the legend for Figure 6. Figures 6 E and F have also been changed for improvement. Regretfully, we did not obtain pseudo-light field images (H&E) alone, so we cannot add them. We would like to show the morphology of the lesions by these pseudo-light field images (H&E).
6. Comment #6: In section 4.2 of the methods, it is mentioned that 5 C57BL/6 animals were used. For which studies were these animals used? My understanding on the design and findings presented in this study is that animals used were control (untreated) apoE-/- and MG-treated apoE-/- mice. Is this not correct?
Responses: We made a shameful mistake. In this manuscript, we did not use C57BL/6 mice. The words have been deleted in Section 4.2. We really appreciate Referee 1.
7. Comment #7: While most studies were performed in 0.3% and 0.4% treatments, in section 2.5 biochemical and molecular analysis were performed only with 0.4% MG. Why?
Responses: Biochemical and molecular analysis were conducted for elucidation of the mechanisms. We considered that the high-dosage group (0.4%) should show a clear response. In addition, there was also an economic reason.
8. Comment #8: Given that biochemical and histopathological changes were detected by MG treatment, how do the authors explain that they did not observe any differences in atherogenic markers?
Responses: Since real-time PCR analysis of atherosclerotic lesions revealed no statistical differences in atherogenic markers (Timp1, Mmp9, Lox1, Cxcl16, and Klf5) between the control and 0.4% MG-treated groups, we considered that MG exhibited anti-atherosclerosis independent of these atherogenic markers. Therefore, next we analyzed macrophage markers in the lesions. In Section 2.5 Biochemical and molecular analyses, we have added the description “We considered that MG exhibited anti-atherosclerosis independent of these atherogenic markers. Next, we analyzed macrophage markers in the lesions.” In the Discussion section (Section 3), we already described “The expression of other atherogenic biomarkers such as Timp1, Mmp9, Cxcl16 and Klf5 was also similar between groups, as analyzed by real-time PCR, indicating a different mechanism operating in the MG-induced reduction of atherosclerosis development in Apoe-/- mice.”
9. Comment #9: Given the effect of MG on body weight and lipid metabolism, and since adipose tissue releases inflammatory factors, could the observed anti-inflammatory effects be due to the reduced body weight and fat of MG-treated mice?
Responses: As suggested by Referee 1, it was shown that increased visceral adiposity induced an unfavorable inflammatory cytokine profile, such as TNFa (Dis. Model. Mech., 2: 231, 2009). By contrast, adiponectin (a protective adipokine) decreases in obesity (Dis. Model. Mech., 2: 231, 2009). One possible reason for the observed anti-atherogenicity by MG administration might be the reduction in body weights (decreased visceral adiposity). This description has been added in the Discussion section (Section 3).
10. Comment #10: While the effect of MG on TH1 cytokines has been well documented, as discussed on page 14, how do the authors explain that in their study no significant effect was seen on IFNg, Tnfa and Il1b inflammatory genes?
Responses: In the present study, mRNA levels of IFNg, TNFa and IL1b showed a decreasing trend in the 0.4% MG-treated group compared to the control group, but was not statistically significant. In animal experiments, data are usually accompanied by large variations in case of strong enough reaction, so that there may not be statistical significance. Since it was recently reported that MG extract markedly reduced TNFa, IL-6 and IL1b in vitro study (Virus Res., 240: 180, 2017; Biomed. Pharmacother., 111: 705, 2019), the observed reduction in TH1 cytokines might be biologically significant. However, further investigation will be required on this point.
11. Comment #11: Since from the analysis of the 2 anti-inflammatory cytokines (Il-4 and Il-13) only Il-13 showed to be altered by MG treatment, do the authors have any evidence on the effect of MG on additional anti-inflammatory markers (eg NFK-b)?
Responses: On the suggestion of Reviewer 1, we learned that it has been reported that the genes for the majority of the pro‐inflammatory proteins are regulated by nuclear transcription factor NF‐κB (J Biol Regul. Homeost. Agents, 23: 141, 2009; J. Physiol. Pharmacol., 64: 409, 2013). In fact, MG has shown to inhibit NF‐κB (Int. Immunopharmacol., 52: 156, 2017). Since NF‐κB response in atherosclerotic lesions of mice treated with MG is very interesting, we would like to analyze the NF‐κB signaling pathway in our next study. Thank you very much for your helpful suggestion.
12. Comment #12: Given the controversial role of LPL on atherogenesis, do the authors have any evidence on additional lipid metabolism enzymes (eg. PPARg, fatty acid synthase) in MG-treated mice?
Responses: MG has been shown to act as a PPAR agonist (Biosci. Biotechnol. Biochem., 77: 2430, 2013; J. Agric. Food Chem., 63: 8399, 2015). In addition, MG has also been reported to inhibit fatty acid synthase in vitro (Mol. Cancer, 13: 138, 2014; Plos One, 7: e33376, 2012). These alterations by MG may be a responsible for anti-atherogenesis. We really appreciate the suggestion by Review 1. Descriptions regarding PPAR and FAS have been added to the Discussion section (Section 3).
13. Comment #13: Could the authors provide a scheme/figure of their overall hypothesis regarding the underlying atheroprotective mechanism of MG based on their finding on the effect of MG on apoE-/- mice.
Responses: A schematic presentation of MG-induced anti-atherosclerosis has been added to Figure 8 and the Conclusion section (Section 5). Thank you very much for your suggestion.
Reviewer 2 Report
Shibata et al., in the manuscript entitled “Crude α-Mangostin Suppresses the Development of Atherosclerotic Lesions in Apoe-deficient Mice by a Possible M2 Macrophage-Medicated Mechanism” investigated the anti-atherosclerotic potential of 0.3% and 0.4% mangosteen extracts in ApoE knockout mice fed with normal chow diet for 17 weeks. Mangosteen extract treatment led to reduced body weight, decreased plasma cholesterol levels and attenuated atherosclerotic lesion formation. In addition, Mangosteen extract-treated mice displayed an increased number of M2 macrophages in the lesion. Overall, the manuscript is well written, “material and methods” section is adequately detailed, and results are clearly presented. However, I have the following concerns:
Minor Comments:
1. There are many typographical and grammatical errors. Even the title of the manuscript is not correct (Macrophage-Medicated Mechanism - Macrophage-Mediated Mechanism).
2. The font type and size are not uniform throughout the manuscript.
Major comments:
1. The authors have studied spontaneous development of atherosclerosis in ApoE-/- mice with a regular chow diet for 17 weeks, and they were 24 weeks old at the time of sacrifice. As per our experience and others (PMID 22701627; 24386260), total cholesterol in 6 months old ApoE-/- mice never reach above 450 mg/dL with chow diet and % area of atherosclerotic plaque is way low than reported in the present study. ApoE-/- mice usually have a plaque area of 15% after 3-4 months of the Western diet.
2. Resolution of histological and immunostaining images are not good. It would have been better if the authors have shown original immunostaining images (macrophage markers) rather than algorithmic images by Image Analysis software.
3. The authors have found reduced atherosclerosis in mangosteen extract-treated mice compared to controls without any difference in the number of CD68+ macrophages in lesions. However, the progression of atherosclerosis occurs with accumulation lipid-laden macrophages in the arterial wall. They need to discuss this in the manuscript.
4. Furthermore, the authors have observed reduced plasma cholesterol levels in extract-treated mice compared to control mice with regular chow diet, which clearly indicate improved metabolism in extract-treated mice. They should have investigated liver enzymes involved in fatty acid transport and synthesis.
Author Response
Responses to Reviewer 2
Minor Comments
1. Comment 1: There are many typographical and grammatical errors. Even the title of the manuscript is not correct (Macrophage-Medicated Mechanism- Macrophage-Mediated Mechanism).
Responses: We made shameful mistake in the title. We really appreciate Reviewer 2. The present manuscript has been further edited by an English editing service for Int J Mol Sci.
2. Comment 2: The font type and size are not uniform throughout the manuscript.
Responses: We have corrected font type and size in the manuscript.
Major Comments
1. Comment 1: The authors have studied spontaneous development of atherosclerosis in ApoE-/-mice with a regular chow diet for 17 weeks, and they were 24 weeks old at the time of sacrifice. As per our experience and others (PMID 22701627; 24386260), total cholesterol in 6 months old ApoE-/- mice never reach above 450 mg/dL with chow diet and % area of atherosclerotic plaque is way low than reported in the present study. ApoE-/- mice usually have a plaque area of 15% after 3-4 months of the Western diet.
Responses: We have read and checked blood cholesterol levels in the references (PMID 22701627; 24386260) indicated by Reviewer 2. Actually, total cholesterol levels in the present study are higher than these references. However, our background data on total cholesterol levels in Apoe-/- mice (14-week-old to 24-week-old) fed a basal diet show 400-800 mg/dl. Although we do not know the reason for the differences, considerable differences are in the basal diet constituents and methodology of the cholesterol measurements. Furthermore, our background data in a plaque area of Apoe-/- mice (14-week old to 24-week old) fed a basal diet show approximately 15%. The observed higher total cholesterol levels are associated with higher plaque area. However, to obtain international standard measurement levels, we should consider this point. Substantial total cholesterol values have been deleted in Section 2.2 Biochemistry. Thank you very much.
2. Comment 2: Resolution of histological and immunostaining images are not good. It would have been better if the authors have shown original immunostaining images (macrophage markers) rather than algorithmic images by Image Analysis software.
Responses: Although the resolution of original histological (Figure 4) and immunostaining (Figure 6) was high quality, since these images were incorporated into the text with an inappropriate method, the resolution was unexpectedly decreased. We have improved the resolutions. Photographs E and F in Figure 6 have also been changed for improvement. Thank you very much.
3. Comment 3: The authors have found reduced atherosclerosis in mangosteen extract-treated mice compared to controls without any difference in the number of CD68+ macrophages in lesions. However, the progression of atherosclerosis occurs with accumulation lipid-laden macrophages in the arterial wall. They need to discuss this in the manuscript.
Responses: On Reviewer 2 suggestion, M1 macrophage accumulation is a hallmark of the progression of atherosclerosis. Accumulation of cholesterol lipids can direct macrophages toward an M1 proinflammatory phenotype (Nat. Rev. Cardiol., 12: 10, 2015). However, since there were no apparent differences in numbers of M1 populations between control and MG-treated mice, the anti-atherogenic effect by MG might be strong enough. In fact, treatment with MG did not strongly decrease M1 polarizing cytokines. It may be improved by alteration of the administration route or in combination with anti-inflammatory drugs. The descriptions have been added to the Discussion section (Section 3).
4. Comment 4: Furthermore, the authors have observed reduced plasma cholesterol levels in extract-treated mice compared to control mice with regular chow diet, which clearly indicate improved metabolism in extract-treated mice. They should have investigated liver enzymes involved in fatty acid transport and synthesis.
Responses: Fatty acid synthase (FAS) and fatty acid transporter also play crucial roles in lipid metabolism by the liver. FAS is a key enzyme required for de novo synthesis of fatty acids, and lipid homeostasis needs fine-tuning of fatty acid transporter (Prostaglandins Leukot Essent Fatty Acids., 82: 149, 2010). Aberrations in lipid metabolism by fatty acid transporter also elicit a pathological state (lifestyle-related disease) (Prostaglandins Leukot Essent Fatty Acids., 82: 149, 2010). The descriptions regarding this have been added to the Discussion section (Section 3).

Round 2
Reviewer 2 Report
Shibata et al., has improved the manuscript significantly, however, there are still pending concerns as mentioned below.
Major comments:
1. The authors need to provide original immunostaining images (macrophage markers) rather than analysis software-generated images.
2. They are strongly suggested to perform the additional experiments to investigate the role of mangosteen-extract treatment in alteration of murine liver enzymes involved in fatty acid transport and synthesis; otherwise, the manuscript is mechanistically weak.
Author Response
Responses to Reviewer 2
Comment #1: The authors need to provide original immunostaining images (macrophage markers) rather than analysis software-generated images.
Responses: Since atherosclerosis has strong autofluorescence in the arterial wall and lipid core, it is difficult to quantitate using a regular fluorescent microscope. Therefore, in order to conduct quantitative analysis of macrophage subtypes (M1 and M2), we hired a Mantra Quantitative Pathology Workstation. Prior to the quantitative analysis, autofluorescence normalization was conducted on unstained sections by the Mantra workstation with inForm Image Analysis software (PerkinElmer). Therefore, we strongly depended on the software-generated images. Regrettably, we did not obtain an individual immunostaining image alone due to inexperience in using this system, so we cannot add them.
Comment #2: They are strongly suggested to perform the additional experiments to investigate the role of mangosteen-extract treatment in alteration of murine liver enzymes involved in fatty acid transport and synthesis; otherwise, the manuscript is mechanistically weak.
Responses: We have added data for Fatty acid synthase (Fasn), Fatty acid transporter (Fatp), HMG-CoA synthase and HMG-CoA reductase in liver. Data are shown in Figure 5C. Although there was no statistical significance in the mRNA levels of Fasn between groups, the levels of Fatp were significantly suppressed in the MG-treated mice compared to control mice (upper panel, Figure 5C). HMG-CoA synthase executes HMG-CoA synthesis from acetoacetyl-CoA. The mRNA levels of HMG-CoA synthase were markedly suppressed in the MG-treated mice, whereas the levels of HMG-CoA reductase, a cholesterol biosynthetic enzyme from HMG-CoA, were significantly elevated (lower panel, Figure 5C). The HMG-CoA reductase levels were elevated in the MG-treated mice, possibly due to a compensatory elevation by decreased HMG-CoA associated with HMG-CoA synthase inhibition. In addition, the observed attenuation of hypercholesterolemia and hepatic steatosis in MG-treated mice in the present study may be due to the suppression of Fatp and HMG-CoA synthase. Thank you very much for your scientific advice. Descriptions regarding the hepatic enzymes have been added in the Abstract, the Results (2.5 Biochemical and molecular analyses), the Discussion, the Materials and Methods (4.6 Cholesterol lipoprotein profiles…..), the References (no.57-60), figure legends in Figure 5C and Figure 8. We really appreciated the Referee 2.
Round 3
Reviewer 2 Report
The authors have satisfactorily addressed the concerns.